# Interested but Uncertain: Carbon Markets and Data Sharing among U.S. Crop Farmers

Guang Han [1,2] and Meredith T. Niles [3,4,*]

1   Department of Rural Development, College of Humanities & Social Development, Nanjing Agricultural University, Nanjing 210095, China; guanghan@njau.edu.cn
2   China Resources & Environment and Development Academy, Nanjing Agricultural University, No. 1 Weigang, Nanjing 210095, China
3   Department of Nutrition and Food Sciences, University of Vermont, Burlington, VT 05405, USA
4   Gund Institute for Environment, University of Vermont, Burlington, VT 05405, USA
*   Correspondence: mtniles@uvm.edu

**Abstract:** The potential for farmers and agriculture to sequester carbon and contribute to global climate change goals is widely discussed. However, there is currently low participation in agricultural carbon markets and a limited understanding of farmer perceptions and willingness to participate. Furthermore, farmers' concerns regarding data privacy may complicate participation in agricultural carbon markets, which necessitates farmer data sharing with multiple entities. This study aims to address research gaps by assessing farmers' willingness to participate in agricultural carbon markets, identifying the determinants of farmers' willingness regarding carbon markets participation, and exploring how farmers' concerns for data privacy relate to potential participation in agricultural carbon markets. Data were collected through a multistate survey of 246 farmers and analyzed using descriptive statistics, factor analysis, and multinomial regression models. We find that the majority of farmers (71.8%) are aware of carbon markets and would like to sell carbon credits, but they express high uncertainty about carbon market information, policies, markets, and cost impacts. Just over half of farmers indicated they would share their data for education, developing tools and models, and improving markets and supply chains. Farmers who wanted to participate in carbon markets were more likely to have higher farm revenues, more likely to share their data overall, more likely to share their data with private organizations, and more likely to change farming practices and had more positive perceptions of the impact of carbon markets on farm profitability. In conclusion, farmers have a general interest in carbon market participation, but more information is needed to address their uncertainties and concerns.

**Keywords:** carbon trading; data privacy; farmer behavior; farmer adoption; payment for ecosystem services

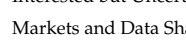


## 1. Introduction

Agriculture is widely recognized for its contribution to greenhouse gas emissions globally and in individual countries, where the contribution varies depending on the level of industrialization [1]. In the United States (U.S.), agriculture contributed roughly 11% of greenhouse gas emissions in 2020, with agricultural soil management contributing the largest source of agriculture emissions [2] and row crop agriculture contributing about 5% of agricultural emissions in the U.S. and Europe [3]. Simultaneously, agriculture has the potential to sequester carbon in soils, with varying levels of impact depending on the soil type, practice implementation, and length of practice change [4]. For both reasons, agriculture has become an increasing area of focus for multinational and national greenhouse gas reduction goals [5]. However, despite this growing focus, many countries do not mandate agricultural emission reductions [6]. Increasingly, though, the private sector marketplace has embraced agriculture in carbon markets or incentive programs,

where farmers are paid to implement greenhouse gas reductions or sequestration. National and state governments have also explored carbon markets as opportunities for farmers to contribute to mitigation goals while also being paid for their reductions [7]. According to the World Bank, there were 70 carbon pricing initiatives globally in 2022, covering 23.17% of global GHG emissions [8]. In the U.S., there are only three current mandatory carbon pricing initiatives (emissions trading schemes), all of which are at the state level. However, as of 2021, there were at least 11 voluntary agricultural carbon credit programs in pilot or action [9], demonstrating that the carbon marketplace for agriculture in the U.S. is largely voluntary. However, despite the burgeoning agricultural carbon marketplace [10], there is little research for understanding farmers' perceptions, concerns, and willingness to participate in agricultural carbon markets. To contribute to this gap, this study aims to assess farmers' willingness to participate in agricultural carbon markets, identify factors determining farmers' willingness regarding carbon markets participation, and explore how farmers' concerns for data privacy relate to potential participation in agricultural carbon markets. We therefore surveyed U.S. row crop (i.e., wheat, barley, corn grain, corn silage, cotton, and soybeans) farmers to understand their perceptions and concerns related to carbon markets, their willingness and drivers for participating in these markets, and their data-sharing preferences, which may be critical for agricultural carbon market monitoring and verification.

### 1.1. Agricultural Carbon Markets Structure

Many farmers have a long history of implementing conservation practices on their farms that confer environmental benefits and ecosystem services, including carbon sequestration and greenhouse gas mitigation [11,12]. In many countries, including the U.S., farmers can be compensated for the implementation of such conservation practices through voluntary participation in government programs at local, state, and national levels. Other farmers may also be incentivized or economically rewarded through private companies, where there is a growing number of agricultural carbon market programs or conservation practice incentive schemes [10]. Similar to government and industry incentive programs, agricultural carbon markets can vary in size and scope, as well as the standards and verification in place within a given program. Carbon markets support the buying and selling of credits, which signify reductions in carbon emissions or the sequestration of carbon. Buyers of carbon credits seek emission reductions either for regulatory or voluntary purposes, and they are either unable to do so themselves or find it economically beneficial to pay for the implementation of greenhouse gas emissions in other parts of the economy. Sellers of credits, including farmers and others (such as foresters), implement changes on their land that (ideally) result in greenhouse gas reductions or sequestration [10].

Schulte Moore and Jordahl [13] defined the carbon market as "a market in which a supply of carbon offset credits is sold to companies that use them to meet their voluntary or regulatory GHG emissions goals or requirements" (p. 10), and a carbon credit is the "unit certified by a carbon credit program or standard that can be traded in carbon markets, representing one metric ton of carbon dioxide equivalent" (p. 10). Unlike government and industry incentive programs, carbon markets include a variety of additional complexities for individual farmers to navigate, including an understanding of the trading process, carbon pricing, the certainty of emission reductions/sequestration of a given practice, and potential buyers. The quality of credits generated can vary based on many factors including additionality (e.g., how "additional" they are, and that they would not have happened otherwise), whether they are permanent (e.g., for a given long time horizon), and verifiability (guaranteed to have occurred) [10]. The carbon market provides farmers with the opportunity to obtain economic benefits by sequestering carbon into soils and plants [14–16]. However, existing carbon markets run by private companies operate trading programs for farmers very differently. Some markets pay farmers based on acreage, with a condition of a minimized size, others pay by carbon credit estimation, and some markets require farmers to implement certain specific carbon sequestration practices [17]. Even

within a marketplace, farmers and carbon credit buyers often operate on different temporal and spatial scales, which can further introduce complexity in the marketplace and often result in a project developer serving as an intermediary [18]. A project developer may play multiple roles in a carbon trading context, but he or she primarily serves to find and initiate projects, often aggregating farmers together and coordinating between suppliers and buyers, including with regard to standards and verification [19].

### 1.2. Data Sharing Requirements for the Carbon Markets and Farmer Willingness to Share

Data infrastructure plays a crucial role in building the transparency, trust, and integrity of carbon markets [20]. A mature carbon market is driven by five types of data: Production data, emissions data, technical data, management data, and economic data [21]. From production and emission perspectives, agriculture provides both sources and sinks of carbon [22]. Farming practices such as different planting and tillage operations, applications of manure or fertilizers, choosing different pesticide programs, irrigation methods, harvesting, and residue management are strongly associated with carbon sequestration and emissions [23]. Therefore, from a technical perspective, site-specific data of farm operations need to be inputted into carbon accounting models to precisely estimate carbon sequestration [24].

Data sharing may be crucially important for agricultural carbon markets, especially as buyers of carbon credits seek high-quality credits that are additional, permanent, and verifiable. For example, some existing private sector carbon markets operating in the U.S. (such as Bayer, Indigo Ag, and Nutrien) do require farmers to share their current and/or historical farm data to be eligible for the trading programs. Farmer data sharing related to management and practice implementation can help achieve these multiple outcomes for high-quality carbon credit generation. Given the strong linkage between farmer data sharing and carbon markets, it is reasonable to assume a strong association between farmers' data sharing willingness and intention to participate in carbon markets. Simultaneously, equipment and management has been outfitted with new technologies that enable "smart farming"—for example, through GPS, sensors, and a veritable rate application of inputs [25]. Many companies have targeted farmer data for their own product development, weather and climate information, and yield and market predictions. However, despite the growth in data sharing needs, there is a limited understanding of farmers' willingness to share data and of their perceptions of the practice. Furthermore, there is even less understanding of how data-sharing preferences relate to carbon market perceptions or participation.

Existing evidence suggests that the majority of farmers, on average, are not willing to share farm data with other organizations and people. For example, Castle et al. [26] found that only 44% of farmers would share data with a university, 43% would share with their local co-op, 37% would share with relatives, and 20% would share with equipment dealers or manufacturers [26]. Relatedly, Turland and Slade [27] found that farmers were most willing to share their data with university researchers and least likely to share with the government. Willingness to share data is affected by farmers' understanding of terms and conditions for data sharing, farmers' trust in the third party, and whether a third party will profit from data sharing. Financial incentives also had significant positive impacts on data sharing through increasing the potential participation in a big data platform [27], which may be relevant for carbon market participation if farmers are being compensated.

### 1.3. Empirical Studies on Farmers' Carbon Market Participation and Data Sharing

Existing research exploring farmers' interests and perceptions of agricultural carbon markets is limited, especially in high-income countries, where our work focuses, despite the growing political and industry interest in carbon markets. Little attention has been given to understanding farmer perceptions and preferences for carbon markets; instead, assumptions that farmers would participate at adequate carbon prices have been common [28]. A significant portion of the research for understanding farmers' preferences and perceptions of carbon credits and markets has been conducted in Australia, where government pro-

grams have been in place for a decade to encourage low-carbon farming and agricultural emission reductions [29]. Farmers primarily cited policy and price uncertainty, as well as a lack of information and perceived high costs for carbon farming, as key barriers to participation in the schemes [29]. Similarly, Dumbrell et al. [30] found that Australian farmers were less likely to participate in carbon farming contracts with higher policy and price uncertainty, as well as when there were uncertain impacts on productivity and profitability from the implementation of carbon farming practices. Fleming et al. [14] further argued that framing carbon markets on their own, as financial opportunities, are not productive, since the majority of farmers they interviewed saw no financial incentives. Instead, carbon market participation, in combination with another potential program such as marketing their products as carbon-neutral or carbon-friendly, was viewed as more promising among farmers [14]. In the U.S., farmers perceived that different types of management changes would incur varying producer costs and benefits, and the majority of producers felt that carbon sequestration would imply a cost to their operations. Perhaps unsurprisingly, farmers were more likely to participate in carbon programs when the expected revenue from such programs increased, but adoption would likely be limited at low carbon prices [28].

Other research has determined the non-financial aspects of market design and farm management that may influence participation and perceptions. Rochecouste et al. [31] demonstrated that the structural components of the Australian carbon market policies themselves may prevent farmers from participating. In particular, additionality and permanence requirements were especially concerning [31]. The technical capacity and understanding of the potential for carbon storage on farms may be another barrier to market participation, as Mattila et al. [32] found that most Finnish farmers who developed carbon farming plans had no knowledge of their farms' existing carbon stock and/or were unaware of carbon balance concepts. Furthermore, the types of practices that farmers most wanted to implement (e.g., cover crops, nutrient amendments, and grassland management) had relatively low carbon storage benefits, which may suggest a mismatch between farmer preferences and carbon sequestration potential at a level to be marketable [32].

In low-income countries, carbon markets are often tied to agroforestry efforts associated with the Reduction of Emissions from Deforestation and Degradation (REDD) implementation and aim to achieve a triple goal of poverty alleviation, food security, and climate change mitigation [33]. Smallholder farmers in low-income countries often lack knowledge that would enable participation in the marketplace [18] and may be marginalized in decision-making processes establishing project requirements or benefits [34]. Furthermore, an analysis of projects has demonstrated that participation in carbon markets resulted in significant increases in labor [35], especially for women, raising important questions about gender equity within carbon markets [34]. The role of project developers or program managers in offset efforts in low-income countries also presents additional power dynamics, which can erode social relations between farmers, managers, and communities enrolled in carbon markets [36].

Given the growth in the volume of available agricultural carbon market opportunities, as well as the political and industry interest in such markets, this research aimed to better understand farmers' preferences and potential participation in such markets in United States agriculture. Furthermore, we aimed to better link farmer preferences and concerns for data sharing with carbon market participation and perceptions, as most carbon markets fundamentally require some aspect of on-farm data sharing for participation. Specifically, we ask the following questions:

1. What factors correlate with farmer interest in carbon market participation?
2. How do farmers perceive carbon markets, including their opportunities and challenges, and how does this influence their potential participation?
3. What kind of data, and for whom, are farmers willing to share? How do data sharing perceptions influence carbon market participation?

## 2. Methods

### 2.1. Data Collection

To address the research questions, we surveyed row crop farmers from a stratified random sample of U.S. farmers across 27 major states for row crops production, including Iowa, Illinois, Indiana, Kansas, Michigan, Missouri, Nebraska, Ohio, Wisconsin, Delaware, Maryland, New Hampshire, New York, Pennsylvania, Virginia, Vermont, Idaho, Washington, Alabama, Florida, Georgia, Kentucky, Louisiana, North Carolina, South Carolina, Oklahoma, and Texas.

Our sampling frame is a database provided by Farm Market ID (now DTN), which is a commercial agricultural data service provider and has been confidently used by scholars for research purposes [37,38]. We selected farmers who operate at least 10 acres of row crops (to primarily include full-time farmers), including barley, corn grain, corn silage, cotton, soybeans, and wheat. Sample stratification is based on the total number of row crop farms in each state, with reference to the Census of Agriculture 2017 [39]. The number of farms selected for each state was chosen in proportion to the amount of row crop acreage in those states. For example, if Alabama farms represented 2.8% of the total number of farms from all row crop agriculture in these states, 2.8% of the total sample came from Alabama. In total, we drew 20,000 farms from the database and electronically surveyed the farmers from February to April 2021 through Qualtrics. Five rounds of e-mail reminders were sent during the data collection period. The electronic survey successfully reached out to 1842 farmers (recipients who opened the survey email). We received 420 responses, resulting in a 22.6% survey cooperation rate (responses among those who opened the email) and a 2.1% response rate (responses among all emails sent) (American Association for Public Opinion Research 2009). Of the responses, we excluded responses that had fewer than 5% completion, and 246 cases were deemed valid for subsequent analyses (*n* = 246).

### 2.2. Variable Selection and Scale Creation

Farmer demographic variables including farmer age, years of farming, formal education, gender, and race/ethnicity as well as farm revenue are considered within our analysis. In addition, we include multiple variables that capture farmers' knowledge and experiences with carbon markets, data sharing and with whom, and current conservation practice adoption (Table A1 in Appendix A).

For multiple questions where we expected that perceptions would be similar across question types, we conducted confirmatory factor analyses to test whether multiple questions had factor loadings greater than 0.40 and eigenvalues greater than 1.00 [40] to facilitate aggregation into a single scale. As a result, we generated seven different scales, which we confirmed for internal validity using a Cronbach alpha, where all scales achieved an alpha higher than 0.70, generally regarded as having high validity [41] (Table 1). Scales related to carbon markets included a *marketpolicyscale*, which captures farmer policy and market perceptions, and a *carbonpracticescale*, which describes farm-related practices and perceptions related to carbon markets. We generated four separate scales related to data sharing including a *datasharingscale* (capturing farmers' willingness to share data for different purposes) and three scales that include the sharing of specific kinds of farm data with different entities (*publicdatascale*, *privatedatascale*, *govdatascale*). Finally, as we expected that farmers' perceptions of carbon markets may also be related to their adoption of different conservation agriculture practices that may have varying carbon benefits, we include a conpracticescale, which captures the adoption of eight different conservation agriculture practices.

**Table 1.** Generation of scales utilized in the analysis, including factor loadings and eigenvalues from confirmatory factor analyses and Cronbach alpha internal validity calculations.

| Scale Name | Question | Scale | Factor Loadings | Eigenvalue | Alpha |
|---|---|---|---|---|---|
| *marketpolicyscale* | Scale variable of three questions: There is not enough information about carbon markets | 1 = strongly disagree, disagree; 2 = uncertain/do not know; 3 = agree, strongly agree | 0.669 | 1.91 | 0.703 |
| | There is too much policy uncertainty regarding carbon markets | | 0.869 | | |
| | There is too much uncertainty in carbon prices | | 0.842 | | |
| *carbonpracticescale* | Scale variable of three questions: I would adjust my farming practices to put more carbon in the soil if carbon markets pay me | | 0.860 | 2.09 | 0.778 |
| | Cover crops can enhance carbon sequestration in the soil | | 0.788 | | |
| | Participating in carbon markets would improve farm profitability | | 0.855 | | |
| *datasharingscale* | Scale variable of four questions: I would share my data for the purpose of developing tools and models | 1 = strongly disagree; 2 = disagree; 2.5 = uncertain or do not know; 3 = agree; 4 = strongly agree | 0.957 | 3.54 | 0.957 |
| | I would share my data for the purpose of crop breeding | | 0.928 | | |
| | I would share my data for the purpose of improving the market and supply chain | | 0.927 | | |
| | I would share my data for the purpose of extension and education | | 0.952 | | |
| *publicdatascale* | Scale variable of willingness to share the following kinds of data with public organizations (e.g., universities, extension, non-profits): | 0 = no, 1 = yes | | 8.93 | 0.969 |
| | cash crop harvesting techniques | | 0.853 | | |
| | cash crop planting technique | | 0.828 | | |
| | cash crop tillage practices | | 0.864 | | |
| | cash crop yield | | 0.804 | | |
| | cover crop biomass | | 0.836 | | |
| | cover crop decomposition rates and nitrogen release | | 0.804 | | |
| | cover crop management practices | | 0.897 | | |
| | crop diseases | | 0.906 | | |
| | pests | | 0.897 | | |
| | production inputs | | 0.872 | | |
| | soil properties | | 0.903 | | |
| | weeds | | 0.884 | | |
| *privatedatascale* | Scale variable of willingness to share the following kinds of data with private organizations (e.g., technology providers): | 0 = no, 1 = yes | | 9.08 | 0.971 |
| | cash crop harvesting techniques | | 0.861 | | |
| | cash crop planting technique | | 0.877 | | |
| | cash crop tillage practices | | 0.774 | | |
| | cash crop yield | | 0.839 | | |
| | cover crop biomass | | 0.876 | | |
| | cover crop decomposition rates and nitrogen release | | 0.864 | | |
| | cover crop management practices | | 0.909 | | |
| | crop diseases | | 0.894 | | |
| | pests | | 0.900 | | |
| | production inputs | | 0.853 | | |
| | soil properties | | 0.901 | | |
| | weeds | | 0.885 | | |
| *govdatascale* | Scale variable of willingness to share the following kinds of data with government organizations (e.g., USDA, state agencies): | 0 = no, 1 = yes | | 9.85 | 0.971 |
| | cash crop harvesting techniques | | 0.881 | | |
| | cash crop planting technique | | 0.870 | | |
| | cash crop tillage practices | | 0.889 | | |
| | cash crop yield | | 0.862 | | |
| | cover crop biomass | | 0.891 | | |
| | cover crop decomposition rates and nitrogen release | | 0.899 | | |
| | cover crop management practices | | 0.925 | | |
| | crop diseases | | 0.956 | | |
| | pests | | 0.929 | | |
| | production inputs | | 0.908 | | |
| | soil properties | | 0.944 | | |
| | weeds | | 0.917 | | |

**Table 1.** *Cont.*

| Scale Name | Question | Scale | Factor Loadings | Eigenvalue | Alpha |
|---|---|---|---|---|---|
| | Scale variable of the current adoption of conservation practices including: | | | 2.83 | 0.709 |
| | Conservation tillage (reduced tillage leaving some residue on the soil surface) | | 0.652 | | |
| | Contour farming (plant and/or till perpendicular to field slopes) | 0 = no, 1 = yes | 0.494 | | |
| *conpracticescale* | Filter/buffer strips | | 0.622 | | |
| | Grassed waterways | | 0.692 | | |
| | No-till (continuous) | | 0.470 | | |
| | No-till (with rotational tillage) | | 0.517 | | |
| | Soil testing for nutrient management | | 0.718 | | |
| | Soil health testing (with soil biological and physical indicators) | | 0.416 | | |

### 2.3. Statistical Analysis and Model Specification

To identify the determinants of farmers' willingness regarding carbon market participation, we utilize a multinomial logit model to capture the factors that may predict these three outcomes. Our primary outcome of interest was farmer interest in carbon market participation, collected through the statement, "I would like to sell carbon credits" (*cma_sell_3*). The variable had three outcomes: strongly disagree/disagree (Y = 1), uncertain/I do not know (Y = 2), and agree/strongly agree (Y = 3). We chose "uncertain/I do not know (Y = 2)" as the reference group. Explanatory variables are consistent for four groups: (1) personal characteristics including *age, fmyear, edu, male,* and *revenue*; (2) carbon market perceptions including *marketknow, cma_familar_3, cma_cost_3, carbonpracticescale,* and *marketpolicyscale*; (3) data sharing preferences including *datasharingscale, publicdatascale, privatedatascale,* and *govdatascale*; (4) conservation practices, which are measured by the *conpracticescale*. The entire statistical analysis was conducted in Stata 17.0 [42]. To check for multicollinearity, we utilized a variance inflation factor (VIF) value following the model (Table A2). All VIF values were below 10, a generally accepted cut-off point for multicollinearity [43,44].

The model specification is:

$$\text{logit}\left(\frac{p(Y=i|X)}{p(Y=2|X)}\right) = \ln\frac{p(Y=i|X)}{p(Y=2|X)}$$
$$= \beta_0 + \beta_{i1}\,age + \beta_{i2}\,fmyear + \beta_{i3}\,edu + \beta_{i4}\,male + \beta_{i5}\,revenue$$
$$+\beta_{i6}\,marketknow + \beta_{i7}\,cma\_familar\_3 + \beta_{i8}\,cma\_cost\_3$$
$$+\beta_{i9}\,carbonpracticescale + \beta_{i10}\,marketpolicyscale + \beta_{i11}datasharingscale$$
$$+\beta_{i12}\,publicdatascale + \beta_{i13}\,privatedatascale$$
$$+\beta_{i14}govdatascale + \beta_{i15}\,conpracticescale$$

where $p(Y = 2 | X)$, or the reference group, represents the probability of the occurrence of the event *cma_sell_3* = 2 given the vector X. $p(Y = i | X)$ represents the probability of the occurrence of the event *cma_sell_3* = i given the vector X.

## 3. Results

### 3.1. Respondent and Farm Demographics

On average, farmers were 55.2 years old (range 25–79, SE = 1.02) and had been farming for 34 years (SE = 1.10). A total of 91% of respondents identified as male, with 8% identifying as female and 1% identifying as another gender. Among these respondents, 95.6% identified as non-Hispanic white, while 4.4% identified with other racial identities and/or Hispanic ethnicity (SE = 2.0%). Forty-eight percent of respondents had a college degree or higher formal education.

The average farm size among respondents was 1838 acres (Table 2), and the top five most prevalent crops grown included corn (85%), soy (82%), wheat (34%), pasture/hay/forage (30%), and corn silage (18%). The average farm net revenue was less than

USD 100,000 for 39% of respondents, more than USD 100,000 but less than USD 500,000 for 35% of respondents, and more than USD 500,000 for 26% of respondents. Among the other conservation practices included in the survey, 53% of farmers had adopted at least one of eight practices, including soil testing for nutrient management (83%), grassed waterways (68%), conservation tillage (67%), filter/buffer strips (50%), soil health testing (45%), rotational no-till (40%), continuous no-till (38%), and contour farming (34%). Table 2 summarizes our sample's characteristics compared to the average U.S. farm and farmers, according to the 2017 Census of Agriculture. Our sample is comparable to the average U.S. farm characteristics for race/ethnicity and age but is over-represented by male and more experienced farmers. Our average farm size is significantly larger than the average U.S. farm size in 2017, likely for two reasons. First, we explicitly excluded farms with less than 10 acres in our sampling, which naturally increases the average farm size. Second, we limited our sample size to five row crops (barley, corn grain, corn silage, cotton, soybeans, and wheat), which have larger farm sizes on average than the U.S. average. For example, in 2017, farms growing corn were, on average, 725 acres [45].

**Table 2.** Sample characteristics compared to the 2017 Census of Agriculture.

| Characteristics | Sample | Census of Agriculture 2017 |
|---|---|---|
| Age (years) | 55.2 | 57.5 |
| Male | 91% | 64% |
| White | 95.6% | 95.4% |
| Farming experience: 11 years or more | 91.4% | 73% |
| Farm operation size (acres) | 1838 | 441 |

### 3.2. Carbon Market Knowledge and Perceptions

Overall, the majority of farms (71.8%) had heard of carbon markets but were not currently participating, while 19.7% had never heard of carbon markets, 6.3% of farmers were currently participating in a carbon market program, and 2.1% had previously participated in a market. The majority of farmers (54.7%) agreed that they would like to sell carbon credits, while 31.2% were uncertain or did not know, and 14.2% disagreed.

Farmers expressed a variety of perceptions related to carbon market policy, economics, and farm behaviors. The majority of farmers expressed high uncertainty about information, policies, markets, and costs (Figure 1). The majority of farmers (72.2%) agreed that there is not enough carbon market information and that there is too much carbon market (67.4%) and carbon price uncertainty (62.9%). There was also uncertainty or confusion about whether carbon markets would improve farm profitability (39.9% uncertain/do not know and 13.8% disagree), whether farming for carbon credits would increase farming costs (41.4% uncertain/do not know and 37.2% agree), and whether farmers were familiar with the trading processes of carbon markets (31.4% uncertain/do not know, 43.6% disagree). However, despite these uncertainties and disagreements, the majority of farmers (59.6%) indicated they would change farming practices to put more carbon in the soil through carbon market payments.

### 3.3. Data Sharing Perceptions and Behaviors

We found that the majority of farmers were willing to share data for the purposes of extension and education (57.1%), developing tools and models (53.6%), and improving the market and supply chains (51.4%), while slightly less than half (47.9%) would share for the purposes of crop breeding (Figure 2). Between 15 and 20% of all farmers indicated uncertainty about sharing data for any of these purposes. We asked farmers about their willingness to share certain kinds of farm data with four different entities (farmers, public organizations such as universities or extensions, private organizations such as technology providers, and government organizations such as the USDA or state agencies) (Figure 3). We find that, across all data types, farmers, overall, are most willing to share data with other farmers (64.8%), public organizations (45.4%), and private organizations (38.0%) and least

willing to share with government organizations (29.0%). While there was some variability about the kinds of data farmers were willing to share (for example, least likely to share yield data), data sharing preferences were generally more about with whom farmers would share, rather than what farmers would share.

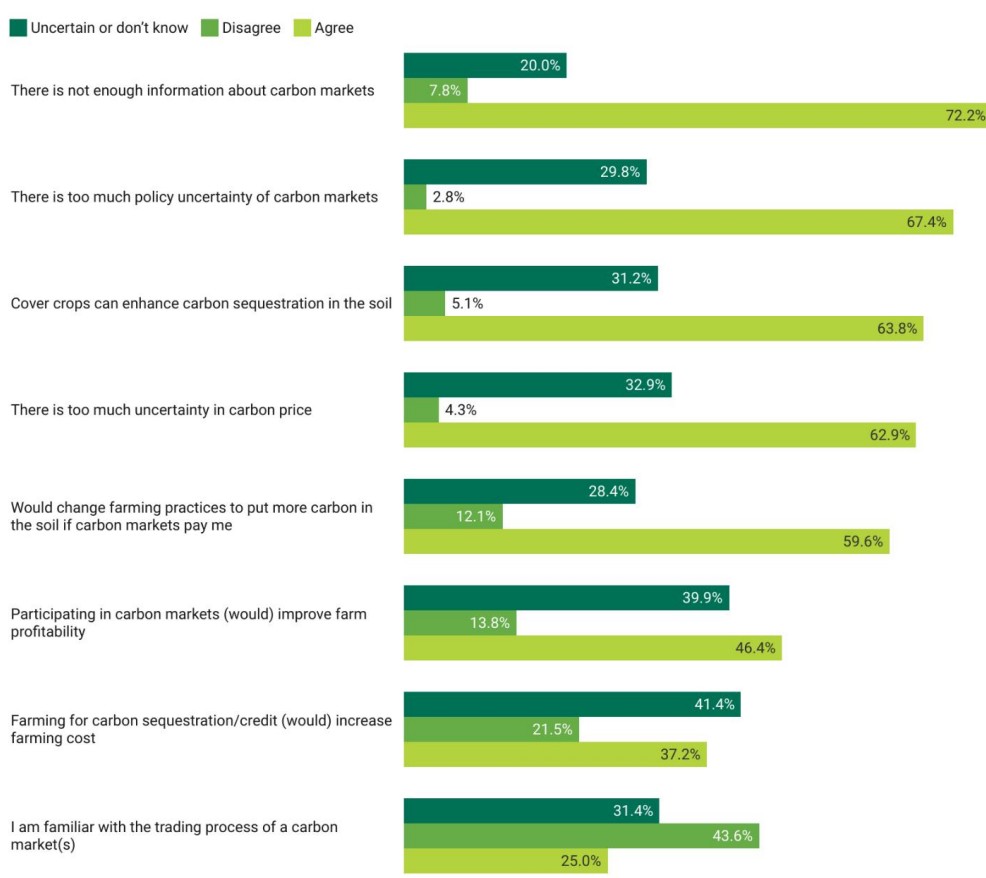

**Figure 1.** Level of farmer agreement related to carbon market components.

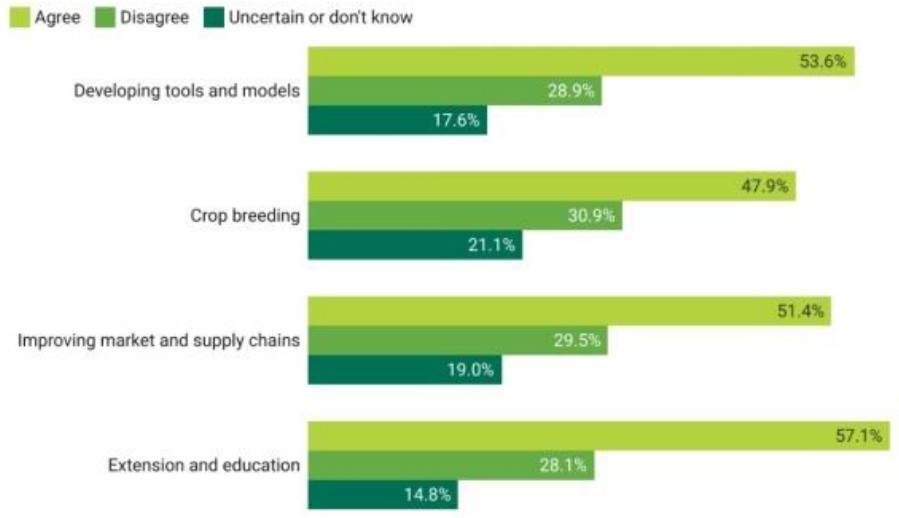

**Figure 2.** Farmer willingness to share data for different purposes.

## Willingness to Share Data with Different Parties

**Figure 3.** Percentage of farmers indicating they would share a certain type of data with different entities.

### 3.4. Carbon Market Participation

A multinomial logit model was utilized to predict factors associated with interest in selling carbon credits in a market. The base outcome of the model is farmers who disagreed that they would like to sell carbon credits ("disagreeing farmers"), as compared with those who agreed they wanted to sell carbon credits ("agreeing farmers") (Table 3) and those who were uncertain/did not know ("uncertain farmers") (Table 4). Only one factor was positively correlated with farmers who agreed they wanted to sell carbon credits or were uncertain as compared to those who disagreed. Farmers who had higher *carbonpracticescale* responses were significantly associated with uncertain farmers ($b = 2.639$, $p = 0.027$) and agreeing farmers ($b = 6.591$, $p \leq 0.001$).

**Table 3.** Multinomial model results predicting farmers who agreed or strongly agreed to sell carbon credits as compared to farmers who disagreed to sell carbon credits. Statistically significant values ($p < 0.10$) are bolded for emphasis.

| Agree/Strongly Agree (Agreeing Farmers) | Coefficient | Standard Error | z-Score | *p*-Value | 95% Confidence Interval | |
| :---: | :---: | :---: | :---: | :---: | :---: | :---: |
| *age* | −0.153 | 0.104 | −1.470 | 0.140 | −0.357 | 0.051 |
| *fmyear* | 0.134 | 0.107 | 1.240 | 0.214 | −0.077 | 0.344 |
| *edu* | 0.591 | 0.587 | 1.010 | 0.314 | −0.560 | 1.742 |
| *male* | −1.602 | 2.325 | −0.690 | 0.491 | −6.159 | 2.955 |
| ***revenue*** | **1.019** | **0.374** | **2.730** | **0.006** | **0.287** | **1.752** |
| *marketknow* | 0.304 | 1.349 | 0.230 | 0.822 | −2.340 | 2.947 |
| ***datasharingscale*** | **1.719** | **1.023** | **1.680** | **0.093** | **−0.287** | **3.724** |
| *publicdatascale* | 0.247 | 1.431 | 0.170 | 0.863 | −2.558 | 3.051 |
| ***privatedatascale*** | **3.610** | **1.664** | **2.170** | **0.030** | **0.349** | **6.872** |
| *govdatascale* | 0.096 | 1.677 | 0.060 | 0.954 | −3.190 | 3.383 |
| *conpracticescale* | −1.454 | 1.841 | −0.790 | 0.430 | −5.063 | 2.154 |
| ***carbonpracticescale*** | **6.591** | **1.428** | **4.620** | **0.000** | **3.793** | **9.389** |
| *marketpolicyscale* | −0.223 | 1.247 | −0.180 | 0.858 | −2.667 | 2.220 |
| *cma_familar_3* | 0.213 | 0.787 | 0.270 | 0.786 | −1.329 | 1.756 |
| *cma_cost_3* | −1.223 | 0.853 | −1.430 | 0.152 | −2.894 | 0.448 |

Note. Pseudo $R^2 = 0.5241$.

**Table 4.** Multinomial model results predicting farmers who were uncertain or did not know about selling carbon credits as compared to farmers who disagreed to sell carbon credits. Statistically significant values ($p < 0.10$) are bolded for emphasis.

| Uncertain/Do Not Know (Uncertain Farmers) | Coefficient | Standard Error | z-Score | *p*-Value | 95% Confidence Interval | |
|---|---|---|---|---|---|---|
| *age* | −0.053 | 0.093 | −0.570 | 0.566 | −0.236 | 0.129 |
| *fmyear* | 0.019 | 0.096 | 0.200 | 0.845 | −0.169 | 0.206 |
| *edu* | 0.157 | 0.496 | 0.320 | 0.751 | −0.815 | 1.130 |
| *male* | −2.496 | 1.863 | −1.340 | 0.180 | −6.147 | 1.156 |
| *revenue* | 0.395 | 0.315 | 1.250 | 0.210 | −0.222 | 1.011 |
| *marketknow* | −0.949 | 1.105 | −0.860 | 0.390 | −3.114 | 1.216 |
| *datasharingscale* | 0.587 | 0.837 | 0.700 | 0.483 | −1.052 | 2.227 |
| *publicdatascale* | −0.085 | 1.291 | −0.070 | 0.947 | −2.616 | 2.445 |
| *privatedatascale* | 2.089 | 1.400 | 1.490 | 0.136 | −0.654 | 4.832 |
| *govdatascale* | 0.019 | 1.345 | 0.010 | 0.988 | −2.616 | 2.655 |
| *conpracticescale* | −0.265 | 1.562 | −0.170 | 0.865 | −3.326 | 2.796 |
| ***carbonpracticescale*** | **2.639** | **1.197** | **2.200** | **0.027** | **0.293** | **4.985** |
| *marketpolicyscale* | −0.752 | 1.084 | −0.690 | 0.488 | −2.876 | 1.373 |
| ***cma_familar_3*** | **1.255** | **0.717** | **1.750** | **0.080** | **−0.151** | **2.660** |
| *cma_cost_3* | −1.079 | 0.778 | −1.390 | 0.165 | −2.604 | 0.445 |

Note. Pseudo $R^2$ = 0.5241.

In addition, there were some unique predictors of uncertain and agreeing farmers as compared to disagreeing farmers. Uncertain farmers were more likely to agree they were familiar with the trading process of carbon markets as compared to disagreeing farmers ($b = 1.255$, $p = 0.080$). Agreeing farmers were more likely than disagreeing farmers to have a higher interest in data-sharing for different purposes (*datasharingscale*) ($b = 1.719$, $p = 0.093$), to be more likely to share their data with private organizations ($b = 3.610$, $p = 0.030$), and to have higher farm revenues ($b = 1.019$, $p = 0.006$).

## 4. Discussion

This work demonstrates that the majority of these surveyed crop farmers are aware of carbon markets and are interested in selling carbon credits, but they have high levels of concern or uncertainty about the carbon market, associated policies, and economic implications. This finding suggests that although a market infrastructure is in place for farmers to trade carbon credits, various challenges have prevented farmers from actually participating in the markets. Schulte Moore and Jordahl [13] found a similar trend in Iowa, where most farmers are interested in carbon markets but were hesitant to take action to actually participate in the market [13]. They argue farmers experienced both demand and supply challenges for them to participate in carbon markets. Our findings point to a lack of information and uncertainty about costs, markets, and policies as major barriers. Our work also echoes some earlier findings about price and policy uncertainty concerns being drivers against participation in carbon market schemes [29,30]. Furthermore, we find that roughly 40% of respondents think that farming for carbon sequestration or credit would increase farming costs or disagreed that carbon market participation would improve farm profitability. Farmers with these perceptions were significantly less likely to want to participate in carbon markets. These concerns may be variable and could also depend on the time horizon for the rate of return on yield or other factors, depending on the practice a farmer might implement for carbon market participation. For example, the installation of a methane digester is a high upfront capital cost, which could have a long time horizon in returns on investment, realized through energy sales or carbon market offsets. Comparatively, the implementation of conservation tillage or cover crop adoption, both of which could have mitigation benefits [46], may have short-term yield loss impacts [47] but could have no or increased yield benefits in a few years [48,49], while also providing income through carbon market participation.

Multiple interventions could be implemented to ameliorate policy and price uncertainty, as well as concerns about farm economic impacts or farm profitability. First, improving farmers' understanding of how carbon markets operate and what price and policy guarantees can or do exist may help to alleviate some of these concerns. Indeed, farmers who better understood the carbon market trading processes were less likely to be disagreeing farmers in this assessment. Given that farmers have demonstrated high trust with farm groups and scientists [50] and that farmers are most willing to share data with other farmers and public organizations such as universities and non-profits, farmer-to-farmer workshops and training regarding carbon market opportunities facilitated with scientists or public organizations may be fruitful. Furthermore, as project developers are often intermediaries between farmers and credit buyers, ensuring that project developers are knowledgeable about agricultural systems and/or work in partnership with trusted entities could be crucial for farmer participation. Finally, given the price and market uncertainties that may exist regardless of regulatory or market controls, driving the adoption of carbon farming practices may also be better facilitated through the communication and demonstration of co-benefits for farmers that go beyond the carbon marketplace. For this reason, Dumbrell et al. [30] and Fleming et al. [14] noted that the communication of potential co-benefits to farmers from the adoption of conservation agriculture practices that sequester carbon or reduce emissions may be critical for adoption. This may be especially important for practices that have longer-term financial payoffs or even short-term yield or economic impacts.

Our findings indicate that farmers' willingness to share data varied by different purposes. Farmers are most likely to share their data for the purpose of extension and education, followed by tools and models development and market and supply chain improvement. This may be related to the fact that farmers tend to be more willing to share data if they perceive that it could directly improve productivity and increase profitability [14], and in the U.S. agriculture system, extension is seen as a primary conduit for farmers' technical assistance for productivity and profitability. In contrast, farmers are less willing to share data for crop breeding, which is influenced by the fact that the perceived primary beneficiaries of data sharing strongly influence farmers' intention to share data [51]. Extension and education, tools and models, and the supply chain will directly benefit farmers by improving farm management and operation. However, the plant breeding industry has been largely privatized in the U.S. and other high-income countries [52]. As a result, crop breeding first benefits agribusiness groups that make profits by selling new cultivars of seeds to farmers and then potentially passing the benefits on to farmers by using the new seeds to increase the crop yield or quality.

Farmers are most likely to share data with other farmers, followed by public and private organizations, but farmers are generally reluctant to share data with governments, largely consistent with previous findings [27,53]. Farmers' willingness to share with other farmers may be, in part, related to social exchange theory, which states that "people help others because the benefits they gain from pro-social behavior are expected to outweigh the costs of providing information to others" [54]. Farmer-to-farmer information exchange is a process that is likely to develop co-benefits, which may benefit the entire farming community in the long run [14,30]. Overwhelmingly, farmers are least willing to share their on-farm data with government actors. These findings suggest that government efforts to develop carbon markets may be more successful if carried out in partnership with other farmer-trusted entities such as public and private organizations. As government entities often serve as regulators of agriculture, farmers may be reluctant to share data with government organizations [27]. Instead, there may be a unique role for public entities including farmer organizations, extension agents, and universities in facilitating carbon market participation or building trust with farmers. These entities may play particularly important roles as project developers or liaisons with government actors.

Importantly, though, data sharing preferences were among the only significant factors influencing agreement with wanting to participate in the carbon market. The overall

propensity to share data was weakly associated with greater carbon market participation, but the willingness to share with private organizations, in particular, was strongly associated with market participation interest. This finding supports our earlier assumption regarding the association between data-sharing willingness and carbon market participation in a more specific context. As today's voluntary agricultural carbon markets in the U.S. are primarily run by the private sector (e.g., Agoro Carbon Alliance, Bayer, CIBO, Corteva, Gradable Carbon, etc.) and require farmers to pull their data into their private labs, project developers, independent verifiers, and carbon registries [13], this result may be a function of the current landscape. However, to the extent that this landscape changes, especially if more regulatory and compliance carbon markets or voluntary markets initiated or led through government entities come online, farmers' willingness to participate may be more limited. Such programs might consider public–private or non-profit–public partnerships for developing participation, based on the results of this work.

Finally, our results also indicate the types of row crop farms and farmers that may be most interested in participating in carbon markets. We find no significant effect of age, years of production experience, or formal education on market participation, unlike Jiang and Koo [28], who found that older farmers were less likely to want to participate, and production experience and college education influenced potential participation in rangeland management carbon programs. However, our results suggest that higher-revenue farms and farmers more willing to change their practices and with favorable views of carbon market profitability are more likely to want to sell carbon credits. These findings are important as policymakers and other carbon market developers consider the expansion of farmer participation. Higher-revenue farms are also likely larger farms, which may offer the opportunity to generate a larger number of carbon credits through fewer transactions, but this also has important equity implications for smaller and medium farms, which are more likely to be run by marginalized or underserved producers, including women [55], beginning farmers and ranchers, and farmers identifying as Hispanic or non-white [10].

Our study bears a few limitations that need to be disclosed. We identify our relatively small sample size, with a low response rate, as a potential limitation of this study, which may limit our findings' wider generalization. However, given the exploratory nature of our study, this paper represents an early attempt to discover farmers' willingness, perceptions, and concerns regarding carbon market and data sharing. In this study, we only employed the electronic survey method, which may exclude farmers who do not have access to smartphones or computers. Our future research will consider using multiple survey methods to increase the response rate, including mail surveys, which, using the same database, have yielded much higher response rates in other surveys [56].

## 5. Conclusions

Here, we surveyed U.S. crop farmers in one of the first assessments of their carbon market and data-sharing perceptions, as interest and participation in such schemes are growing. Our results identify that the majority of farmers (71.8%) are interested in carbon markets, but most farmers (more than 60%) also exhibit uncertainty around market and policy specifics. This suggests that as growth in the carbon market industry continues, farmers will require additional specific details about different opportunities, and many will seek technical information. Furthermore, engagement with agricultural communities for carbon market participation should consider how data sharing and data privacy may influence their participation and what assurances farmers may need to share data with different entities, especially government organizations. This study is one of the first attempts to document some of the early perceptions of farmers, as enrollment in agricultural carbon markets is limited. Future research should continue to track farmers' perceptions and willingness to participate in the carbon markets as well as data privacy perceptions. Furthermore, additional research could better assess specific areas of technical assistance, knowledge of carbon market structures, and scenarios for potential participants that could guide policy development.

**Author Contributions:** Conceptualization, M.T.N. and G.H.; methodology, M.T.N. and G.H.; software, M.T.N.; validation, M.T.N. and G.H.; formal analysis, M.T.N.; investigation, G.H.; resources, M.T.N.; data curation, G.H.; writing—original draft preparation, M.T.N. and G.H.; writing—review and editing, M.T.N. and G.H.; visualization, G.H.; supervision, M.T.N.; project administration, M.T.N.; funding acquisition, M.T.N. All authors have read and agreed to the published version of the manuscript.

**Funding:** This research was funded by the USDA NIFA Sustainable Agricultural Systems (SAS) Coordinated Agricultural Projects (CAP) Grant, grant number 2019-68012-29818. This research was also partially supported by the Nanjing Agricultural University Humanities and Social Science Fund, grant number: SKYC2023008; Central Universities Basic Scientific Research Business Fees, grant number: KYCXJC2023006; Jiangsu Province Department of Education General Projects of Philosophy and Social Science Research in Colleges and Universities, grant number: 2023SJYB0057; and Jiangsu Province Department of Agriculture and Rural Affairs Rural Revitalization Soft Science Fund, grant number: 23ASS045.

**Data Availability Statement:** The datasets generated during and/or analyzed during the current study are not publicly available yet but will be available at the completion of the grant. The data are available from the corresponding author upon reasonable request.

**Acknowledgments:** The authors thank Jennifer Thompson, Rob Myers, and Rod Rejesus for their feedback on our survey instrument and results. The authors thank the members of the Niles lab for their helpful feedback and review of an initial draft of this paper.

**Conflicts of Interest:** At the time of publication (but not at the time of data collection, analysis, and original writing), MTN was a consultant to Lindahl Reed on the USDA Partnerships for Climate Smart Commodities program.

## Appendix A

**Table A1.** Variable names, questions, and scales.

| Variable Name | Question | Scale |
|---|---|---|
| *age* | What is your age? | Continuous |
| *fmyear* | How long have you been farming? | Continuous |
| *edu* | What is your highest level of formal education? | 1 = Less than a high school degree; 2 = High school graduate/GED; 3 = Associate degree, apprenticeship or trade experience; 4 = 4-year college degree; 5 = Graduate or professional degree (e.g., JD, MS, PhD); |
| *male* | What is your gender? | 1 = Male, 0 = Female or another gender identity |
| *revenue* | Please select the option that best describes the net revenue from your farm operation in 2019. | 1 ≤ USD 10,000; 2 = USD 10,000–USD 49,999; 3 = USD 50,000–99,999; 4 = USD 100,000–249,999; 5 = USD 250,000–499,999; 6 = USD 500,000–999,999; 7 = USD 1 million and more. |
| *Non-white_Hispanic* | Please specify your race/ethnicity. Please select all that apply to you. White; Hispanic, Latino, or Spanish origin; Black or African American; Asian; Native American or American Indian; Native Hawaiian or Other Pacific Islander; Other, specify | 1 = Any non-white Hispanic identity, 0 = White, non-Hispanic identity |
| *marketknow* | To what extent are you aware or engaged in carbon market programs that pay farmers for carbon sequestration and storage or reducing greenhouse gas emissions? Never heard of these markets; Heard of these markets, but not participating; Have previously participated in a carbon market program; I am currently participating in a carbon market program | 1 = Any respondent who heard of the markets, previously or currently participating, 0 = Never heard of these markets |

**Table A1.** *Cont.*

| Variable Name | Question | Scale |
|---|---|---|
| *cma_sell_3* | I would like to sell carbon credits | |
| *cma_familar_3* | I am familiar with the trading process of a carbon market(s) | |
| *cma_cost_3* | Farming for carbon sequestration/credits (would) increase farming costs | |
| *marketpolicyscale* | Scale variable of three questions: There is not enough information about carbon markets There is too much policy uncertainty regarding carbon markets There is too much uncertainty in carbon prices | 1 = strongly disagree, disagree; 2 = uncertain/do not know; 3 = agree, strongly agree |
| *carbonpracticescale* | Scale variable of three questions: I would adjust my farming practices to put more carbon in the soil if carbon markets pay me Cover crops can enhance carbon sequestration in the soil Participating in carbon markets would improve farm profitability | |
| *datasharingscale* | Scale variable of four questions: I would share my data for the purpose of developing tools and models I would share my data for the purpose of crop breeding I would share my data for the purpose of improving the market and supply chain I would share my data for the purpose of extension and education | 1 = strongly disagree; 2 = disagree; 2.5 = uncertain or do not know; 3 = agree; 4 = strongly agree |
| *publicdatascale* | Scale variable of willingness to share the following kinds of data with public organizations (e.g., universities, extension, non-profits): cash crop harvesting techniques, cash crop tillage practices, cash crop yield, cover crop biomass, cover crop decomposition rates and nitrogen release, cover crop management practices, crop diseases, pests, production inputs, soil properties, weeds. | |
| *privatedatascale* | Scale variable of willingness to share the following kinds of data with private organizations (e.g., technology providers): cash crop harvesting techniques, cash crop tillage practices, cash crop yield, cover crop biomass, cover crop decomposition rates and nitrogen release, cover crop management practices, crop diseases, pests, production inputs, soil properties, weeds. | Farmers indicated 0 = no, 1 = yes for each of the data types. Scale variable ranges from 0 to 1, with incremental fractions. |
| *govdatascale* | Scale variable of willingness to share the following kinds of data with government organizations (e.g., USDA, state agencies): cash crop harvesting techniques, cash crop tillage practices, cash crop yield, cover crop biomass, cover crop decomposition rates and nitrogen release, cover crop management practices, crop diseases, pests, production inputs, soil properties, weeds. | |

**Table A1.** *Cont.*

| Variable Name | Question | Scale |
|---|---|---|
| *conpracticescale* | Scale variable of current adoption of conservation practices including: conservation tillage, contour farming, filter/buffer strips, grassed waterways, continuous no-till, rotational non-till soil testing for nutrient management, soil health testing | Farmers indicated 0 = no, 1 = yes for each of the conservation practice types. Scale variable ranges from 0 to 1, with incremental fractions. |

**Table A2.** Variance Inflation Factor (VIF) values for multicollinearity check.

| Variable | VIF |
|---|---|
| age | 7.31 |
| fmyear | 7 |
| edu | 1.39 |
| male | 1.26 |
| revenue | 1.15 |
| BIPOC | 1.24 |
| marketknow | 1.34 |
| datasharingscale | 1.8 |
| publicdatascale | 2.1 |
| privatedatascale | 1.78 |
| govdatascale | 1.59 |
| conpracticescale | 1.41 |
| carbonpracticescale | 1.47 |
| marketpolicyscale | 1.3 |
| cma_familar_3 | 1.36 |
| cma_cost_3 | 1.16 |
| cc_ever | 1.46 |
| Mean VIF | 2.13 |

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
