# Peer review of "Interested but Uncertain: Carbon Markets and Data Sharing among U.S. Crop Farmers"

_land, doi:10.3390/land12081526_

Round 1
Reviewer 1 Report
The abstract of the contribution lacks the aim of the work and the methods that were used in the work.
The literature review is at a high scientific level. At the end of the review, there are mentioned questions, hypotheses to which the authors want to answer, but I miss the defined bad goal, again.
The entire questionnaire is described in the methodology and correlations are also mentioned, but in the Results chapter at the end I see the model that has R2 in the note. I assume that the authors calculated a regression model. You mention a multinomial logit model, but it is not described in the methodology. I miss this model in the description of the methodology. It would be appropriate to mention it in the Results as well. What "z" stands for and "p" stands for. If you had so many variables in the model, didn't you check for multicollinearity?
Author Response
The abstract of the contribution lacks the aim of the work and the methods that were used in the work.
Response: Thanks for pointing out this issue. We have rewritten parts of the abstract and added the information in the abstract following the reviewer’s suggestions (lines 13 -21).
The literature review is at a high scientific level. At the end of the review, there are mentioned questions, hypotheses to which the authors want to answer, but I miss the defined bad goal, again.
Response: The research goal statement was added to the introduction section (lines 54-58).
The entire questionnaire is described in the methodology and correlations are also mentioned, but in the Results chapter at the end I see the model that has R2 in the note. I assume that the authors calculated a regression model. You mention a multinomial logit model, but it is not described in the methodology. I miss this model in the description of the methodology. It would be appropriate to mention it in the Results as well. What "z" stands for and "p" stands for. If you had so many variables in the model, didn't you check for multicollinearity?
Response: The reviewer is correct. We did run multinomial logit model. Following the reviewer’s suggestions, we added model specifications in the methodology section (lines 249-line 265).
"z" stands for z-score of the beta coefficient, which is the regression coefficient divided by the standard error. "p" stands for significance level. Per the 7th APA publication manual, researchers are encouraged to use “z” to represent a standardized score and use "p" stands for significance level ( American Psychology Association, 2019, p.185). We have updated the tables to explicitly define these.
In addition, we did check for multicollinearity using a variance inflation factor. We have added this additional detail in the methods, as well as an appendix table listing the VIF for all variables, which were all under 10, a generally accepted value for low multicollinearity.
Reference: American Psychology Association. (2019). Publication manual of the American psychology association.
Reviewer 2 Report
Agriculture is the main supplier of carbon dioxide, which contributes to global warming. The study of issues that will reduce carbon dioxide emissions and predict the amount of emissions to stop climate change are very important. The authors of the article conducted theoretical studies on the readiness of American farmers for publicity, providing real data on the economy, and cultivation technologies, and conducted analytical studies on the readiness of farmers to participate in carbon markets.
The title of the article needs to be improved - in the title, the authors indicate row crops, and in lines 201 and 202 they indicate barley and wheat, which are not row crops (better write - agricultural crops). Also, the text of the article and the tables contain the phrases “pastures/hay/fodder and others”, which are also not included in the concept of row crops.
The abstract of the article, as well as the conclusions, do not contain clear digital indicators that must be present.
The article lacks a clear understanding of what a carbon market is and what benefits the farmers who participate there can have. There are also no references to what financial advantages farmers have if they participate in the carbon market, or how they can sell their quotas.
The list of used literature is compiled from new and modern studies.
Author Response
Agriculture is the main supplier of carbon dioxide, which contributes to global warming. The study of issues that will reduce carbon dioxide emissions and predict the amount of emissions to stop climate change are very important. The authors of the article conducted theoretical studies on the readiness of American farmers for publicity, providing real data on the economy, and cultivation technologies, and conducted analytical studies on the readiness of farmers to participate in carbon markets.
Response: We sincerely thank the reviewer’s compliments.
The title of the article needs to be improved - in the title, the authors indicate row crops, and in lines 201 and 202 they indicate barley and wheat, which are not row crops (better write - agricultural crops). Also, the text of the article and the tables contain the phrases “pastures/hay/fodder and others”, which are also not included in the concept of row crops.
Response: Thank you for pointing out this inconsistency, though this may be related to regional differences in how row crops are defined. To make descriptions more consistent, we decided to remove “row crop” from our title, and further refine in the methods what we mean by row crop.
The abstract of the article, as well as the conclusions, do not contain clear digital indicators that must be present.
Response: We added a few descriptive statistics in the abstract (lines 18-19). The Abstract has a length limit and we already added more content required by reviewer #1, so we don’t have more room to add more statistics in the abstract. We did add a few more descriptive statistics in the conclusion section (lines 482-483).
The article lacks a clear understanding of what a carbon market is and what benefits the farmers who participate there can have. There are also no references to what financial advantages farmers have if they participate in the carbon market, or how they can sell their quotas.
Response: Thanks for pointing out this drawback of our manuscript. We have a whole subsection (1.1 Agricultural Carbon Markets Structure) describing how agricultural carbon markets operate. To make it clearer, we added official definitions of the carbon market and carbon credits in the literature section (lines 80-84). We also added three references (Fleming et al., 2019; Cowie et al., 2019; Evans, 2018) to support our argument that the carbon market provides farmers with the opportunity to obtain economic benefits by sequestering carbon into soils and plants (lines 90-91).
References:
Fleming, A.; Stitzlein, C.; Jakku, E.; Fielke, S. Missed Opportunity? Framing Actions around Co-Benefits for Carbon Mitigation in Australian Agriculture. Land Use Policy 2019, 85, 230–238, doi:10.1016/j.landusepol.2019.03.050.
Cowie, A.L.; Waters, C.M.; Garland, F.; Orgill, S.E.; Baumber, A.; Cross, R.; O’Connell, D.; Metternicht, G. Assessing Resilience to Underpin Implementation of Land Degradation Neutrality: A Case Study in the Rangelands of Western New South Wales, Australia. Environmental Science & Policy 2019, 100, 37–46, doi:10.1016/j.envsci.2019.06.002.
Evans, M.C. Effective Incentives for Reforestation: Lessons from Australia’s Carbon Farming Policies. Current Opinion in Environmental Sustainability 2018, 32, 38–45, doi:10.1016/j.cosust.2018.04.002.
The list of used literature is compiled from new and modern studies.
Response: Thank you, given the agricultural carbon markets are still emerging, we did try to cite new and modern literature.
Reviewer 3 Report
A good analysis of carbon market conditions
Author Response
Thank you for your review.
Reviewer 4 Report
Dear Authors,
Thank you for submitting your paper. I have read it with great interest, focusing on the farmers' behavior regarding data sharing. However, I have a few major and minor observations regarding the manuscript.
Firstly, the paper lacks clarity in terms of its novelty and scientific soundness within the existing literature. The literature review fails to clearly highlight the specific research gap that this manuscript aims to address.
Secondly, the argument presented in the paper is mainly descriptive and lacks the depth needed to explain why farmers are hesitant to share their data. The thesis statement provided, "Farmers that wanted to participate in carbon markets were more likely to have higher farm revenues, be more likely to share their data overall, more likely to share their data with private organizations, more likely to change farming practices, and have more positive perceptions of the impact of carbon markets on farm profitability," does not adequately address the reasons behind the correlation between higher farm revenues and participation in carbon markets. Additionally, the statement combines different results, making it confusing for the reader. It would be beneficial to provide more insights into the underlying reasons for these observed behaviors.
Regarding Table 1, I suggest including it in the annexure attached to the article to enhance the readability and flow of the manuscript.
Moreover, please consider italicizing the names of composite variables in the text (Ln 237 and wherever it has appeared) to clearly indicate their special status as terms or variable names.
Lastly, in Ln 243, it appears that you intended to mention the primary variable of interest. Please revise the sentence to provide the correct information.
Overall, I believe addressing these concerns will significantly improve the clarity and impact of your manuscript. I look forward to reviewing the revised version.
Regards,
Author Response
Dear Authors,
Thank you for submitting your paper. I have read it with great interest, focusing on the farmers' behavior regarding data sharing. However, I have a few major and minor observations regarding the manuscript.
Firstly, the paper lacks clarity in terms of its novelty and scientific soundness within the existing literature. The literature review fails to clearly highlight the specific research gap that this manuscript aims to address.
Response: Thanks for pointing out this potential weakness. We have clarified the aims and goals of this study (lines 54-59). The research gap statements were also covered in lines 52-54. The highlights of our contribution to the literature body were stated in lines 486-497.
Secondly, the argument presented in the paper is mainly descriptive and lacks the depth needed to explain why farmers are hesitant to share their data. The thesis statement provided, "Farmers that wanted to participate in carbon markets were more likely to have higher farm revenues, be more likely to share their data overall, more likely to share their data with private organizations, more likely to change farming practices, and have more positive perceptions of the impact of carbon markets on farm profitability," does not adequately address the reasons behind the correlation between higher farm revenues and participation in carbon markets. Additionally, the statement combines different results, making it confusing for the reader. It would be beneficial to provide more insights into the underlying reasons for these observed behaviors.
Response: We appreciate the reviewer’s interest in better understand the “why” behind some of our results. However, we must be careful with going beyond correlation of variables with our analysis, since we can’t make causal inferences with a cross-sectional survey of this nature. As a result, we have chosen to highlight the characteristics of the farmers that are more interested in participating, but can’t elicit why those types of farmers might be interested, beyond ideas based on the previous literature, which we note already.
Regarding Table 1, I suggest including it in the annexure attached to the article to enhance the readability and flow of the manuscript.
Response: Thanks for the reviewer’s suggestion. We moved the original Table 1 to Appendix A (lines 527-528). Now the table was renamed as Table A1. We also rearranged the table numbers given the move of Table 1.
Moreover, please consider italicizing the names of composite variables in the text (Ln 237 and wherever it has appeared) to clearly indicate their special status as terms or variable names.
Response: To make the texts easier to read, we have italicized all the variable names in all tables and texts.
Lastly, in Ln 243, it appears that you intended to mention the primary variable of interest. Please revise the sentence to provide the correct information.
Response: We have thoroughly rewritten the Statistical Analysis and Model Specification part (lines 257-274), we hope the revision would improve the clarification.
Overall, I believe addressing these concerns will significantly improve the clarity and impact of your manuscript. I look forward to reviewing the revised version.
Response: Thanks for the reviewer’s constructive suggestions. We have followed the reviewer’s advice and made the revisions. Hope the quality of our paper is improved and meet the reviewer’s expectation.
Round 2
Reviewer 4 Report
Thanks for acknowledging and addressing the comments.